# Murine Long Noncoding RNA Morrbid Contributes in the Regulation of NRAS Splicing in Hepatocytes In Vitro

**DOI:** 10.3390/ijms21165605

**Published:** 2020-08-05

**Authors:** Anna Fefilova, Pavel Melnikov, Tatiana Prikazchikova, Tatiana Abakumova, Ilya Kurochkin, Pavel V. Mazin, Rustam Ziganshin, Olga Sergeeva, Timofei S. Zatsepin

**Affiliations:** 1Center of Life Sciences, Skolkovo Institute of Science and Technology, 121205 Moscow, Russia; Anna.Fefilova@skolkovotech.ru (A.F.); T.Prikazchikova@skoltech.ru (T.P.); T.Abakumova@skoltech.ru (T.A.); Ilia.Kurochkin@skolkovotech.ru (I.K.); p.mazin@skoltech.ru (P.V.M.); 2Serbsky National Medical Research Center for Psychiatry and Narcology, 119034 Moscow, Russia; proximopm@gmail.com; 3Shemyakin-Ovchinnikov Institute of Bioorganic Chemistry, 117997 Moscow, Russia; Rustam.Ziganshin@gmail.com; 4Department of Chemistry, Lomonosov Moscow State University, 119992 Moscow, Russia

**Keywords:** alternative splicing, long noncoding RNA, liver, nonsense-mediated decay

## Abstract

The coupling of alternative splicing with the nonsense-mediated decay (NMD) pathway maintains quality control of the transcriptome in eukaryotes by eliminating transcripts with premature termination codons (PTC) and fine-tunes gene expression. Long noncoding RNA (lncRNA) can regulate multiple cellular processes, including alternative splicing. Previously, murine Morrbid (myeloid RNA repressor of Bcl2l11 induced death) lncRNA was described as a locus-specific controller of the lifespan of short-living myeloid cells via transcription regulation of the apoptosis-related Bcl2l11 protein. Here, we report that murine Morrbid lncRNA in hepatocytes participates in the regulation of proto-oncogene NRAS (neuroblastoma RAS viral oncogene homolog) splicing, including the formation of the isoform with PTC. We observed a significant increase of the NRAS isoform with PTC in hepatocytes with depleted Morrbid lncRNA. We demonstrated that the NRAS isoform with PTC is degraded via the NMD pathway. This transcript is presented almost only in the nucleus and has a half-life ~four times lower than other NRAS transcripts. Additionally, in UPF1 knockdown hepatocytes (the key NMD factor), we observed a significant increase of the NRAS isoform with PTC. By a modified capture hybridization (CHART) analysis of the protein targets, we uncovered interactions of Morrbid lncRNA with the SFPQ (splicing factor proline and glutamine rich)-NONO (non-POU domain-containing octamer-binding protein) splicing complex. Finally, we propose the regulation mechanism of NRAS splicing in murine hepatocytes by alternative splicing coupled with the NMD pathway with the input of Morrbid lncRNA.

## 1. Introduction

During the last decades, the basic dogma of molecular biology—DNA↔RNA→protein—evolved into a complex network with back loops that regulate gene expression in eukaryotes in a cell- and tissue-specific manner. Advances in high-throughput next-generation sequencing (NGS) dramatically changed genome and transcriptome studies. Based on a big data analysis, thousands of noncoding RNA were discovered and proposed as essential regulators of gene expression [1,2]. Long noncoding RNA (lncRNA) can interact with DNA, RNA or proteins, resulting in the modulation of transcription, chromatin remodeling, formation, targeting and stabilization of functional ribonucleoproteins [3]. Several lncRNA are potential targets and biomarkers in some cancers and cardiovascular and central nervous system (CNS) diseases [4,5,6,7].

Alternative splicing (AS) is among the key processes that control gene expression and perform transcriptome reprogramming in eukaryotes. Alternatively, spliced messenger RNA (mRNA) variants can produce functionally diverse altered protein isoforms with varied activity and/or localization [8]. Alternatively, spliced exons can introduce premature translation termination codons (PTC) in transcripts. Such PTC-containing splice variants are degraded through the nonsense-mediated decay (NMD) pathway [9]. Initially discovered as a quality control (QC) step that helps to remove aberrant splicing transcripts, the NMD pathway can also contribute in the fine-tuning of gene expression via so-called regulated unproductive splicing and translation (RUST) [10]. RUST modulation of transcript levels is achieved by the enhanced or decreased production of PTC-containing splice variants further degraded by NMD without protein synthesis. Lewis et al. [11] estimated that ~30% of alternatively spliced exons introduce PTC. This fact demonstrates the widespread coupling of alternative splicing and NMD for the regulation of gene expression. Recent studies show that some lncRNA are involved in the regulation of alternative splicing [12]. There are two main mechanisms how lncRNA can modulate AS: LINC01133, GOMAFU and LINC-HELLP act as sponges for splicing factors [12], while others like asFGFR2 can participate in RNA-RNA interactions with pre-mRNAs [13]. For example, MALAT1 lncRNA acts as a molecular decoy for SR splicing factors, which decreases their available portion in the nucleus by capturing proteins in the nuclear speckles [14]. NEAT1 and MALAT1 can also influence the phosphorylation of splicing regulatory proteins or act as chromatin remodelers. Another example is lncRNA 51A, a natural antisense transcript to the intron 1 of the sortilin-related receptor 1 (SORL1) gene. LncRNA 51A binds to SORL1 pre-mRNA and masks canonical splicing sites, resulting in an alternatively spliced short protein form [15]. Hence, lncRNAs can modulate transcriptome reprogramming in eukaryotes using several distinct mechanisms.

The present study is focused on the uncovering of moonlight functions of murine Morrbid lncRNA in hepatocytes. Previously, Kotzin et al. demonstrated that Morrbid lncRNA plays an important role in the proliferation and survival of immune cells in mice [16,17,18]. Morrbid knockout mice have reduced levels of eosinophils, neutrophils and Ly6C^hi^ classical monocytes in the peripheral blood and tissues [16]. In the mouse genome, Morrbid is located close to the proapoptotic gene *Bcl2l11* and functions in cis to regulate the *Bcl2l11* promoter through the recruitment of the polycomb repressive complex 2 (PRC2) [16]. Thus, the expression of a long noncoding RNA serves as a locus-specific controller of the life span of short-lived myeloid cells. In preleukemic Hematopoietic Stem and Progenitor Cells (HSPC) depleted of the *Tet2* gene, Morrbid is a key component of the signaling pathway promoting the enhanced survival and reduced apoptosis of *Tet2*-KO cells [18]. Despite the apparent inhibitory role of Morrbid in the regulation of *Bcl2l11* in myeloid cells and HSPC, in CD8+ T cells, Morrbid was shown to be essential for the upregulation of *Bcl2l11* expression in response to Lymphocytic Choriomeningitis Virus (LCMV) infection signaling [17]. In spite of a thorough characterization of Morrbid lncRNA in myeloid cells, nothing is known about its functions in other cells and tissues. Many lncRNA demonstrated tissue-specific expression and functions due to varied cell-specific regulatory networks [19]. Thus, functions of lncRNA Morrbid may significantly differ between immune cells and hepatocytes.

LncRNAs are poorly conserved among vertebrates and rarely have identical functionality even between humans and mice [20]. Initially, murine lncRNA Morrbid was proposed by Kotzin et al. as a homolog of human cancer-related lncRNA MIR4435-2HG [16]. Both lncRNA are located in the second chromosome in reverse orientation, have a similar genomic environment and have conserved fragments in exons. The upregulation of MIR4435-2HG (or human Morrbid) correlates with the progression of human hepatocellular carcinoma (HCC), so MIR4435-2HG was proposed as a novel HCC biomarker [21,22]. However, functions of murine Morrbid in hepatocytes and in HCC development were not shown yet. Here, we found that Morrbid lncRNA participates in the regulation of alternative splicing of the proto-oncogene NRAS mRNA in hepatocytes. We observed that one NRAS splice variant containing a small supplementary exon with PTC was highly enriched in the nuclear RNA population of hepatocytes with Morrbid lncRNA depletion. We found that a low basal level of this NRAS splice variant in the cytosol is determined by its degradation via the NMD pathway. We demonstrated that Morrbid lncRNA interact with splicing complex SFPQ-NONO and can control the production of the NRAS splice variants with PTC through the AS-NMD mechanism.

## 2. Results

### 2.1. Inhibition of Morrbid lncRNA Decreases the Viability and Migration of Murine Hepatocytes

First, we compared Morrbid lncRNA levels in murine cancer (Hepa1-6) and normal (AML12) cell lines and found that Morrbid expression in normal cells was ~2.5-fold higher in comparison to Hepa1-6 (Figure 1A). Then, we studied the subcellular localization of Morrbid lncRNA in hepatocytes by a fluorescence in situ hybridization analysis (FISH) and found that the lncRNA predominantly localizes in the cell nucleus (Figure 1B). To confirm this data, we separated the nuclear and cytoplasmic fractions of AML12 and Hepa1-6 cells and measured the Morrbid lncRNA levels by RT-qPCR. In both cell lines, ~70% of Morrbid lncRNA was detected in the nuclear RNA fraction (Figure 1C).

We designed antisense oligonucleotides (ASO), validated each of them in vitro and used a combination of the four most active ASO that target different exons of Morrbid lncRNA in further studies (Appendix A). We demonstrated that transfection of the ASO mix results in an ~80% decrease of the Morrbid lncRNA expression in AML12 cells in comparison to control ASO (targets the firefly luciferase gene, marked as LUC control) (Appendix A). The efficacy of the lncRNA downregulation after 24 h of ASO treatment was confirmed with the FISH analysis (Appendix A). Then, we estimated the viability of Hepa1-6 and AML12 cells by the resazurin assay after Morrbid lncRNA depletion in optimized conditions. After an initial slight decrease of cancer cells viability (about 20%), they completely recovered at day four after ASO transfection (Figure 1D). In the case of normal AML12 cells, viability gradually decreased following ASO transfection, reaching a 70% reduction at day four (Figure 1D). We estimated the impact of Morrbid lncRNA on the migration ability of AML12 and Hepa1-6 cells using the wound-healing assay. In the case of AML12 cells, the migration rate of hepatocytes with Morrbid knockdown (KD) was significantly reduced in comparison to the control cells (Figure 1E and Appendix A). On the other hand, Hepa1-6 cells depleted in Morrbid showed no changes in migration (Figure 1E and Appendix A). Thus, Morrbid lncRNA is highly represented in a noncancerous hepatocyte cell line and promotes the cell viability and migration of normal liver cells. Unlike its human homolog, murine Morrbid lncRNA was not upregulated in the cancer cell line, and its inhibition did not negatively affect the proliferation or migration of murine cancerous hepatocytes.

### 2.2. Morrbid Depletion Affects the Alternative Splicing of Oncogene NRAS mRNA

In mature short-living myeloid cells, Morrbid lncRNA promotes the H3K27me3 histone modification of the *Bcl2l11* gene promoter through interaction with the polycomb repressive complex 2 (PRC2) [16]. To uncover alternative roles of Morrbid lncRNA in normal hepatocytes, we performed an RNA-seq analysis of AML12 cells with Morrbid KD and defined 1988 genes with significantly altered expressions (*p*-value < 0.05). Among them, 1244 were upregulated and 744 downregulated (two-fold change was used as a threshold). Most of these genes are involved in signal transduction in cancer (MAPK pathway, TGF-β, p53 and NF-ƙB pathways); apoptosis and peroxisome and mitochondrial metabolisms (Appendix A List of genes differentially expressed in Morrbid ASO KD). Thus, Morrbid lncRNA is significant for the functioning of normal hepatocytes and participates in multiple processes that influence the decreased viability and motility in Morrbid KD hepatocytes.

Next, we studied if Morrbid lncRNA depletion affects alternative splicing (AS) in the AML12 cell line. We mapped the RNA-seq data against the splicing graph to detect novel alternative splicing events. We found 84 AS events for 79 genes with significant changes (generalized linear model (GLM), quasi-likelihood ratio test and BH correction; *p* < 0.05): 38 of them represent the differential expression of alternative 5′-donors, 30—cassette exons, 15—3′-acceptors and one—retained intron (Appendix A). Among them, we focused our attention on the proto-oncogene NRAS, which demonstrated a significantly increased inclusion of cassette exons in hepatocytes (Figure 2A). NRAS is a member of the Ras gene family (NRAS, KRAS and HRAS), which is involved in the regulation of cell proliferation and migration by inducing downstream signaling cascades, such as MAPK/Erk and PI3K/Akt [23]. The downregulation of Morrbid leads to the increased incorporation of a 96-nt-long alternative exon into NRAS mRNA (Ensembl ID ENSMUSE00000742446) between the first and second exons. This cassette exon contains a premature stop codon, and its inclusion into mature mRNA can result in the nonsense mediated decay degradation of the NRAS transcript (Appendix A). We therefore named this 96-nt exon the NRAS PTC exon. This transcript is a product of PTC exon inclusion and a potential target for NMD decay. The main NRAS transcript that leads to production of the mature NRAS protein is named as the NRAS-no-PTC transcript or simply the NRAS transcript. To differentiate and compare the expression levels between various NRAS isoforms, we designed a set of primers (Appendix A) targeting either the region shared by all NRAS transcripts (NRAS total), the isoform that lacks the PTC exon (no PTC) or the NRAS PTC transcript (Appendix A). To detect NRAS PTC transcripts, we used primers that amplify the entire PTC exon (primer pair PTC 96nt), primers laying across the junctions with neighboring exons 1 and 2 (primer pair PTC junction) and primers that span across the PTC-exon 2 junction (PTC downstream or PTC down) (Appendix A). Using RT-qPCR, we confirmed that the relative expression of the NRAS PTC transcript to the NRAS total is approximately 3.5-fold higher in cells with depleted Morrbid lncRNA than in control cells (Figure 2B and Appendix A), while the expression level of the isoform lacking the PTC exon is unaffected by Morrbid KD (Figure 2B). Then, we amplified the control DNA (cDNA) using primers laying within exon 1 and exon 2 and spanning across the PTC exon (Figure 2C) and analyzed products on the agarose gel. There are two evident bands (Figure 2C) that correspond to NRAS transcripts with (397nt) and without alternative PTC exons (301nt), as was confirmed by Sanger sequencing (Appendix A). Additionally, we estimated the amount of NRAS pre-mRNA by RT-qPCR and found an increase in the Morrbid KD cells (Figure 2D). Therefore, Morrbid lncRNA downregulation leads to the increased expression of NRAS pre-mRNA and increased portion of the alternative NRAS PTC transcript with a cassette PTC exon.

### 2.3. Morrbid lncRNA Interacts with the SFPQ-NONO Heterodimer

LncRNA can regulate splicing by multiple mechanisms [12]. In order to find functional protein partners of Morrbid lncRNA, we performed a modified capture hybridization analysis of RNA targets (CHART) in AML12 cells followed by a liquid chromatography–mass spectrometry (LC-MS) analysis of proteins crosslinked with Morrbid [24]. We used several biotinylated probes complementary to Morrbid lncRNA exons that cover most of Morrbid-annotated transcripts. Two independent Morrbid-CHART experiments followed by protein separation using Laemmli polyacrylamide gel electrophoresis (PAGE) resulted in the enrichment of specific protein bands (molecular mass ~120 kDa) in comparison to the controls (Appendix A). In-gel trypsinolysis followed by a LC-MS analysis showed that these proteins are SFPQ and NONO (Figure 3A). SFPQ and NONO are members of the Drosophila behavior/human splicing (DBHS) protein family, each containing two RNA-binding motifs. Together, they form the SFPQ-NONO heterodimer that regulates multiple steps of the RNA metabolism, including alternative splicing [25]. To confirm that SFPQ and NONO directly interact with Morrbid lncRNA in cells, we performed a RIP (RNA immunoprecipitation) analysis using antibodies against SFPQ and NONO proteins. Morrbid lncRNA was enriched in SFPQ (12-fold) and NONO (60-fold) fractions in comparison to nonspecific IgG control (Figure 3B). RNA-binding helicase DDX3 was used as an additional control and demonstrated only five-fold enrichment for Morrbid lncRNA. A control RNA did not show any enrichment in SFPQ and NONO protein fractions in comparison to the IgG control. Thus, results of the combined CHART LC-MS and RNA pulldown assays clearly show that Morrbid lncRNA directly and specifically interacts with SFPQ and NONO proteins.

Previously, it was shown that SFPQ and NONO proteins can interact with pre-mRNA regulatory elements and affect alternative splicing [26,27]. In order to verify if the SFPQ-NONO heterodimer can directly interact with NRAS pre-mRNA, we performed an RNA pulldown analysis using biotinylated transcripts of four minigenes. These constructs were: (1) minigene coding a full NRAS PTC exon flanked with 68 nt of exon1 upstream and 137 nt of exon2 downstream, (2) minigene coding an exon1-exon2 fusion with a missing PTC exon and (3) and (4) controlling minigenes that were created by inverting sequences of the first two constructs (Appendix A). A mass-spectrometry analysis of RNA pulldown samples unambiguously confirmed the interaction of SFPQ proteins with the NRAS total minigene (Figure 3A). Therefore, the SFPQ protein interacts directly with both lncRNA Morrbid and NRAS mRNA, while the NONO protein interacts with Morrbid lncRNA.

In order to study if SFPQ is involved in the regulation of NRAS splicing, we depleted SFPQ using the RNA interference (RNAi) technique. We designed and validated six small interfering RNAs (siRNAs) targeting SFPQ mRNA and selected the most efficient ones (Appendix A). The SFPQ protein was not detected by Western blot analysis after six days of siRNA treatment, while some proteins still remained in the cells after four days of knockdown (Appendix A). In SFPQ-depleted cells, the ratio of the NRAS PTC transcript to the NRAS total was increased approximately two-fold in comparison to cells transfected with control siRNA, while the expression of NRAS-no-PTC remained unchanged (Figure 3C). Thus, the SFPQ protein is involved in the regulation of splicing of the NRAS PTC transcript.

### 2.4. NRAS PTC Transcript underGoes NMD Decay in the Cytoplasm

To explore whether the NRAS PTC transcript is indeed degraded in the cytoplasm via the NMD pathway, we separated the nuclear and cytoplasmic fractions of the RNA from the Morrbid KD and control cells. We demonstrated by RT-qPCR that the distribution of both the NRAS total and NRAS-no-PTC transcripts in the cytoplasm versus nucleus was ~3:7. At the same time, the NRAS PTC transcript was almost undetectable in the cytoplasmic fraction (>95% was localized in the nucleus) (Figure 4A and Appendix A). To confirm the NMD nature of NRAS PTC transcript degradation, we measured the half-life of transcripts by the actinomycin D assay. Blocked transcription in AML12 cells by actinomycin D led to a time-dependent gradual decrease of all NRAS transcripts, but the degradation rate of the NRAS PTC transcript was ~four times higher than for the total NRAS and NRAS-no-PTC transcripts. The half-life of the NRAS PTC transcript was 2.6 h, while the half-life of the NRAS transcript with the omitted PTC exon was 10.9 h and NRAS total—11.8 h (Figure 4B). These results support the hypothesis that the NRAS PTC transcript is quickly degraded in the cytosol by the NMD mechanism. For additional proof of the NMD pathway involvement, we used siRNAs to downregulate the key NMD factor UPF1 (siRNA-targeting firefly luciferase gene (Luc siRNA) was used as a control). Downregulation of the UPF1 protein resulted in an increase of the NRAS PTC transcript, while the NRAS transcript lacking the PTC exon remained unchanged in UPF1 KD cells (Figure 4C). This data confirms that the level of the NRAS PTC transcript depends on the NMD factor UPF1 and finally proves the NMD of the NRAS PTC transcript in the cytosol.

## 3. Discussion

The actual amount of specific mRNA relies on the rates of its synthesis and degradation in the cell. Among many regulatory mechanisms, the degradation of some mRNA transcripts is performed by alternative splicing coupled with the nonsense-mediated mRNA decay pathway (AS-NMD). Particularly, AS results in transcript isoforms with PTC followed by the degradation of these mRNA by NMD [28]. Genome-wide studies have shown that 5% to 10% of the *Saccharomyces cerevisiae* [29], *Caenorhabditis elegans* [30] and *Drosophila melanogaster* [31] transcriptomes are changed when NMD is inactivated.

Multiple noncoding RNA are involved in the regulation of alternative splicing and nonsense-mediated mRNA decay pathways. LncRNA may interact with splicing factors; form duplexes with pre-mRNA or perform chromatin remodeling, modulating transcription and splicing. Recently, the lncRNAs NEAT1 and MALAT1 were shown to be colocalized with nuclear speckles containing splicing factor SC35 [32]. Additionally, lncRNA may interact with the target mRNA (for example, half-STAU1-binding site RNAs (1/2-sbsRNAs)) and create a double-stranded transactivation motif that binds to the STAU1-double-stranded (ds)RNA-binding protein and induce mRNA degradation [33].

In this work, we found that murine Morrbid lncRNA is involved in the regulation of the AS-NMD pathway for the proto-oncogene NRAS. Previously, lncRNA Morrbid was identified as a PRC2-dependent inhibitor of the proapoptotic gene *Bcl2l11* in myeloid cells [16]. Here, we demonstrated that the expression of Morrbid lncRNA in normal hepatocytes is higher in comparison with cancer cells. This data contradicts with previous reports on the upregulation of human Morrbid lncRNA in HCC and can be the result of differed lncRNA functions between species. As Morrbid is preferably localized in the nucleus, we used modified antisense oligonucleotides (ASO) to downregulate Morrbid lncRNA in AML12 cells [34]. The knockdown of Morrbid lncRNA by ASO led to the decrease of hepatocyte proliferation and migration rates (Figure 1D,E). This data correlates with the already published function of murine Morrbid lncRNA in neutrophils, eosinophils and classical monocytes, in which Morrbid is crucial for the physiologic control of the lifespan [16]. However, lncRNA functions can vary in different cell types, so we performed an RNA-seq analysis of Morrbid-depleted hepatocytes and found the dysregulation of many transcripts involved in several signaling pathways. A further analysis of the splicing variants in the RNA-seq data of hepatocytes with Morrbid depletion revealed >80 alternative splicing events. We were interested by the accumulated NRAS transcript with a PTC cassette exon (“poison exon”), which could be a target of nonsense-mediated decay based on the Ensembl database annotation (Figure 2A). To confirm this feature, we measured the relative expression of the NRAS PTC transcript in Morrbid KD cells and found its significant upregulation in comparison to the total NRAS (Figure 2B). During the next step, we demonstrated that this NRAS PTC transcript with a cassette exon is a target of NMD, as this transcript is undetectable in the cytosol (Figure 4A). The NRAS PTC transcript shows almost three times the decreased half-life in comparison to the total NRAS in normal hepatocytes, but it is rather stable—2.6 h in comparison with the published data of the NMD targeted transcripts [35] (Figure 4B). Additionally, in hepatocytes with depleted UPF1, the NRAS PTC transcript is upregulated (Figure 4C), which additionally proves that this NRAS transcript is a target for NMD. Previously, Barbie et al. showed that alternative splicing is an important mechanism for RAS regulation [36]. The RAS protein family member HRAS has a cassette exon containing PTC, which, upon inclusion, leads to a quick NMD degradation in the cytosol. The incorporation of this HRAS NMD exon was favored in response to genotoxic stress in a p53-dependent manner, suggesting a stress response mechanism for the regulation of HRAS cellular levels. In the case of NRAS, it was shown previously that different isoforms possess different oncogenic activity. Human melanoma cells produce five NRAS isoforms, expressed at different levels, resulting in a varied activation of downstream pathways, levels of phosphorylated *p*-Erk and *p*-Akt and resistance to anticancer drugs [37,38].

To identify how Morrbid lncRNA may be involved in the regulation of NRAS splicing, we performed protein pull-down assays and found that Morrbid interacts with the SFPQ-NONO heterodimer (Figure 3B), which is a splicing-related protein complex [39]. SFPQ and NONO bind as a complex to the conserved stem loop within snRNA U5 and under splicing conditions and assemble U5 snRNA together with other U5-specific factors like the U5/U5/U6 tri-SNP [40]. Despite direct binding, DBHS proteins do not represent the essential components of the splicing machinery; rather, they function as alternative splicing regulators [25]. SFPQ and NONO regulate both the inclusion (for example, the N30 exon of NMHC mRNA and exon7 of the SMN2 gene) and exclusion (for example, exon4 of PPT, Tau and CD45) of multiple cassette exons [26,27,41,42,43]. Thus, we propose that SFPQ and NONO interact with lncRNA Morrbid to mediate the exclusion of the cassette exon, with a premature stop codon in NRAS pre-mRNA. This result correlates with published data showing that SFPQ binds to the stem-loop structure of the 5′ splice site of microtubule-associated Tau pre-mRNA, thus promoting the exclusion of one of the microtubule-binding repeat regions [27]. Another example includes SFPQ binding to the ESS1 (exonic splicing silencer) regulatory element in transmembrane tyrosine phosphatase CD45 pre-mRNA. This interaction results in the skipping of three exons and production of a catalytically inactive CD45 protein [26]. Several known lncRNAs bind to SFPQ and NONO to perform regulatory functions. The most well-known lncRNA partner of DHBS proteins is NEAT1. The SFPQ-NONO-NEAT1 complex forms a scaffold core of the subnuclear structures called paraspeckles [44]. Another example is lncRNA VL30, which regulates the transcription of several genes (GAGE6 and Rab23) via an RNA protein decoy-like mechanism [45,46].

To prove the direct interaction of the SFPQ-NONO complex with NRAS pre-mRNA, we performed a protein pull-down assay using the coding parts of NRAS transcripts and found that the SFPQ-NONO complex interacts only with the main NRAS mRNA transcript but not with the PTC isoform (Figure 3A). Interestingly, SFPQ often binds to the consensus sequence UGGAGAGGAAC on pre-mRNA in order to promote splicing, with the middle nucleotides (AGAGGA) representing patterns that interact with SFPQ more frequently [40]. We found an AGAGGA sequence within exon 2 of NRAS mRNA (Appendix A). We propose that a SFPQ binding site may be located at the junction of exon1 and exon2 of the total NRAS mRNA, while the NRAS NMD transcript may form secondary structures that prevent SFPQ binding.

We propose the mechanism of the regulation of NRAS splicing variants by alternative splicing coupled with the nonsense-mediated mRNA decay pathway under control of the Morrbid lncRNA in hepatocytes (Figure 5). Under normal conditions, Morrbid lncRNA can interact with the SFPQ-NONO heterodimer and participate in interactions of the SFPQ-NONO complex with NRAS pre-mRNA to improve the maturation of NRAS mRNA. In the case of Morrbid depletion, the efficacy of the SFPQ-NONO heterodimer binding to NRAS pre-mRNA decreases, resulting in alternative splicing of NRAS mRNA. These events promote the inclusion of the “poison exon” with PTC, leading to the degradation of such transcript by the NMD pathway. Thus, Morrbid lncRNA contributes to the correct splicing of the main NRAS mRNA isoform.

In conclusion, we performed an alternative splicing analysis in murine hepatocytes with Morrbid lncRNA depletion in vitro and found the upregulation of NRAS pre-mRNA and the oncogene isoform with PTC and proved its degradation via the nonsense-mediated decay pathway. We confirmed the direct interactions of Morrbid lncRNA with splicing complex SFPQ-NONO and proposed the mechanism for the regulation of NRAS splicing in murine hepatocytes by alternative splicing coupled with the NMD pathway with the input of Morrbid lncRNA.

## 4. Materials and Methods

### 4.1. Antisense Oligonucleotides (ASO) and Small Interfering RNA (siRNA)

To perform the downregulation of Morrbid lncRNA expression, we designed 13 chemically modified gapmer ASO (Appendix A). First, the secondary structure of Morrbid transcript variant 3 (NCBI Reference Sequence NR_028591.1) was simulated using ViennaRNA RNAfold WebServer [47]. Then, ASO binding sites in Morrbid lncRNA were chosen based on the accessibility within the secondary structure. We aligned RNase H cleavage sites at loops, multiloops or internal loops. ASO were synthesized as gapmers to improve exonucleolytic stability and binding to the RNA target. Ten central nucleotides were phosphorothioate 2′-deoxynucleotides to maintain RNase H catalytic activity, while 5′- and 3′- flank four to six nucleotides—2′-OMe phosphorothioate nucleotides. The control ASO was designed to target the Firefly Luciferase gene. Additionally, we designed six chemically modified siRNAs (Appendix A) targeting the murine SFPQ sequence (NCBI Reference Sequence NP_076092.1). The siRNAs were selected to avoid off-target activity based on several known criteria [48,49,50]. In order to estimate the off-target binding capacity, siRNA 19-mer sequences were screened against the RefSeq mRNA database. Specifically, siRNAs were filtered based on the number/positions of the mismatches in the seed region, mismatches in the nonseed region and mismatches in the cleavage site position. Then, chosen candidate sequences were checked for the presence of known miRNA motifs and immune stimulatory sequence motifs [49] that should be avoided. Resulting siRNAs were further chemically modified with 2′-OMe pyrimidine nucleotides and single 3′-internucleotide phosphorothioates in order to reduce the immune response and off-target effects and increase stability against nucleases [49,51]. The control siRNA targets the Firefly Luciferase gene. siRNA against UPF1 were previously designed using the same protocol and tested in our lab. siRNA UPF1 target both murine UPF1 transcript variant 1 and transcript variant 2 (NCBI Reference Sequences NM_001122829.2 and NM_030680.3) (Appendix A).

### 4.2. Cell Culture and Transfection

Murine liver cell lines—normal AML12 (ATCC number: CRL-2254) and hepatoma Hepa 1-6 (provided by Prof. O. Dontsova, Moscow State University, Moscow, Russia)—were cultured in Dulbecco’s Modified Eagle Medium: Nutrient Mixture F-12 (DMEM/F12) medium (Thermo Fisher Scientific, Waltham, MA, USA) and supplemented with 10% fetal bovine serum (Thermo Scientific Gibco) and 1% penicillin-streptomycin (Thermo Fisher Scientific, Waltham, MA, USA) at 37 °C and 5% CO_2_.

First, siRNA or ASO (Appendix A) were premixed with Lipofectmine RNAi or Lipofectamine 2000 (Thermo Fisher Scientific, Waltham, MA, USA), respectively, and opti-MEM reduced the serum medium (Thermo Fisher Scientific, Waltham, MA, USA) according to the manufacturer’s protocol. To perform cell transfection, the premix was added to adherent cells to achieve a 10-nM final concentration of ASO/siRNA. Cells were incubated for 1 day to measure mRNA knockdown efficacy, for 2 days to perform the knockdown of Morrbid or for 6 days to perform SFPQ and UPF1 protein knockdowns, with repeated transfections at days 2 and 4. Efficacy of RNA downregulation was quantified by RT-qPCR and efficacy of protein knockdown by Western blot analysis.

### 4.3. RNA Isolation and RT-qPCR

Total RNA was isolated using TRIzol (Thermo Fisher Scientific, Waltham, MA, USA), according to the manufacturer’s instructions. In order to remove any residual DNA from samples, isolated total RNA (0.5 μg) was further treated with 1U of DNase I (Thermo Fisher Scientific, Waltham, MA, USA), supplied with a RiboLock RNase Inhibitor (0.4 U/μL). cDNA was synthesized using a Maxima First Strand cDNA Synthesis Kit (Thermo Fisher Scientific, Waltham, MA, USA), followed by qPCR using a qPCRmix-HS LowROX Kit (Evrogen, Moscow, Russia), according to the manufacturer’s protocols. qPCR was performed using primers listed in Appendix A. Positions of NRAS primers are shown in Appendix A. Gapdh was used as a reference gene for the RT-qPCR analysis.

### 4.4. Separation of Nuclear and Cytoplasmic Cell Fractions

For separation of the cytoplasmic and nuclear fractions, cells were resuspended in a low-salt buffer (20-mM HEPES (pH 7.9), 10% glycerol, 1.5-mM MgCl_2_, 0.05% NP−40 and 0.4-U/μL RiboLock RNase inhibitor). After intensive mixing, cells were kept on ice for 5 min and centrifuged at 1500× *g* for 5 min at 4 °C. Cytoplasmic fraction was collected as a supernatant. Remaining pellet was then resuspended in high-salt buffer (20-mM HEPES (pH 7.9), 10% glycerol, 1.5-mM MgCl_2_, 0.05% NP−40, 0.4-U/μL RiboLock RNase inhibitor (40 U/μL) and 0.5-M NaCl), kept on ice for 10–15 min and centrifuged at 12,000× *g* for 5 min at 4 °C to isolate supernatant with nuclear fraction. Then, RNA was isolated both from cytoplasmic and nuclear fractions using TRIzol, as described above. Gapdh was used as a reference for the cytoplasmic fraction, and snRNA U6 was used as a reference for nuclear fraction.

### 4.5. Fluorescent in Situ Hybridization (FISH)

AML12 cells were cultivated on poly-L-lysine-coated microscopy glasses. First, cells were washed twice with phosphate-buffered saline (PBS) and fixed with 4% paraformaldehyde (PFA) (Sigma Aldrich, St. Louis, MO, USA) at room temperature (RT) for 20 min. Then, cells were washed with PBS twice for 10 min and permeabilized for 10 min with 0.5% Triton X-100 in PBS. Then, cells were washed with PBS for 10 min, followed by two washes in “FA wash” buffer (40% formamide, 2.4×SSC (36-mM sodium chloride and 36-mM sodium citrate in water). Negative-control glasses were additionally incubated with RNase A at 37 °C for 1 h (200 ug/mL of RNase A in 2X SSC buffer (30-mM sodium chloride and 30-mM sodium citrate in water). Then, all glasses with cells were placed in the hybridization chamber and stained with 40 ng of a Cy5 oligonucleotide probe TTGCCTGGAAAGTCACTTTG-Cy5 in the hybridization buffer (0.4% BSA (Sigma Aldrich, St. Louis, MO, USA), 36-mM sodium chloride and 36-mM sodium citrate in water and 44% formamide (*v/v*) supplied with a tRNA+ssRNA mix (5.5 μg/μL) overnight at 37 °C. After that, cells were washed twice with FA wash buffer, preheated up to 37 °C for 15 min and then washed once with PBS at RT for 10 min. Thereafter, the cells were stained with 4′,6-diamidino-2-phenylindole (DAPI) (Thermo Fisher Scientific, Waltham, MA, USA) (300 nM in DMF) for 2 min, washed with PBS and studied by confocal microscopy. Confocal microscopy was performed using a Nikon A1+MP confocal imaging system using a Plan Apo 20×/0.75 Dic N objective (numerical aperture = 0.75, Nikon, Japan), Apo LWD 40×/1.15 S water immersion objective (numerical aperture = 0.15, Nikon, Japan) and Apo tirf 60×/1.49 DIC oil immersion objective (numerical aperture = 0.49, Nikon, Japan). Images were scanned sequentially using 561-nm diode lasers in combination with a DM561-nm dichroic beam splitter.

### 4.6. Cell Viability Assay

AML12 and Hepa1-6 cells were plated into 48-well plates in triplicates, ~25×10^3^ cells per well, and transfected with Morrbid or control ASO (final concentration 10 nM). Viability of cells was measured at 24h, 36h, 48h and 72h timepoints after initial transfection. Measurements were done using 1% (*w/v*) resazurin sodium salt stock solution in PBS, which was further diluted 250-fold with PBS and added to adherent cells (preliminary washed twice with PBS), followed by 1-h incubation at 37 °C. Then, the fluorescent signal was measured using a Varioscan Microplate reader with a 540/590-nm (ex/em) filter set (Thermo Fisher Scientific, Waltham, MA, USA). To calculate the results, the ratio between Morrbid knockdown and control knockdown was calculated for each timepoint and then normalized to the initial signal at the zero timepoint.

### 4.7. Wound-Healing Assay

AML12 cells were cultured in 6-well plates until confluence reached 70–80%, and then, cells were transfected with Morrbid or control ASO (final concentration 10 nM). Upon transfection, several wounds with a width of approximately 1.5 mm were introduced in cell monolayers using a pipette tip, and a first image of the wounds was made at the zero timepoint. To trace the wound closure, pictures were taken at 24 h, 48 h and 72 h after wound introductions in 6 replicas per each sample. Total wounding area was measured using ImageJ software, as was described previously [52]. The results were normalized to the corresponding wound areas at the zero timepoint.

### 4.8. RNA-seq Data Processing and Analysis

For transcriptome analysis, we used ~6 × 10^6^ AML12 cells per sample after 48 h of ASO-mediated Morrbid KD or control LUC KD; 4 replicates per experiment were used. Total RNA was extracted using a TRIzol reagent (Thermo Fisher Scientific, Waltham, MA, USA) according to the manufacturer’s instructions. Six micrograms of total RNA (quantified using a NanoDrop OneC Spectrophotometer ((Thermo Fisher Scientific, Waltham, MA, USA) was fragmented using conditions optimized to result in average 200-nt RNA fragments: incubation for 7 min at 95 °C in RNA fragmentation buffer (100-mM Tris (pH 8.0) and 2-mM MgCl_2_). Fragmented RNA was purified by precipitation in 100% ethanol with a 1/10 volume of 3-M sodium acetate, and 1 µg of RNA (measured using a NanoDrop™ OneC Spectrophotometer (Thermo Fisher Scientific, Waltham, MA, USA) was used for an rRNA depletion reaction using a NEBNext rRNA Depletion Kit (NEB E6310L, New England Biolabs, Ipswich, MA, USA) according to the manufacturer’s protocol. Then, the RNA solution was diluted with a 1/10 volume of 3-M sodium acetate, RNA precipitated with ethanol, and 300 ng of RNA (measured using a NanoDrop™ OneC Spectrophotometer (Thermo Fisher Scientific, Waltham, MA, USA) were used for the sequencing library preparation with a NEBNext Ultra II Directional RNA Library Prep Kit for Illumina (NEB 7760, New England Biolabs, Ipswich, MA, USA), according to the manufacturer’s protocol, and the resulting double-stranded cDNA was purified using AMPure XP magnetic beads (A63881, Beckman Coulter, Brea, CA, USA). Efficient concentrations of libraries were determined using RT-qPCR. Library quality (length distribution and the absence of primer dimers) was assessed on a Bioanalyzer2100 (Agilent Technologies, Santa Clara, CA, USA).

Libraries were pooled in equal amounts and sequenced using a HiSeq4000 (Illumina, San Diego, USA) instrument in 50-nt single-read mode, according to the manufacturer’s protocol. Conversion to fastq format and demultiplexing was performed using bcl2fastq2 software (Illumina, San Diego, USA). Morrbid lncRNA KD and control KD samples were sequenced, returning a variable number of paired reads.

For mapping those samples, genome annotations were obtained from Ensembl. Paired-end reads were mapped using STAR v2.5.3a [53] with default settings, except for the following one: –quantMode GeneCounts. The resulting gene counts were further processed with R package DESeq2 [54], where it was further normalized using the RLE method. Additionally, in order to take into account unwanted data variations, we introduced additional variables, obtained by sva package [55], that capture the unwanted variations into a design matrix. The DESeq2 package were used for performing a differential expression analysis based on the Wald test. We defined genes as differentially expressed if they passed the thresholds: FDR < 0.05 and |log2foldChange| > 0.5.

For the alternative splicing analysis, we mapped all RNA-seq reads using the HISAT2 (v2.1.0) program [56], with the --no-softclip parameter on the mouse genome (assembly GRCm38) using the splice site coordinates from the Ensembl annotation. Then, the data was processed using the SAJR pipeline [57]. Briefly, each gene was split into the regions between two adjacent splice sites (segments)—based on the exon/intron coordinates from the Ensembl annotation. Alternative segments (that are included in some transcripts and excluded from others) were classified into three types according to the combinations of types of splice sites that define their borders: (i) cassette exons are segments that start from acceptor sites and end with donor sites, (ii) alternative acceptor (donor) segments are segments that both start and end with acceptor (donor) sites and (iii) retained introns are segments that start at a donor site and end with an acceptor site. For each segment and each sample, we calculated the number of inclusion reads (i.e., reads that overlap exons by at least one nucleotide) and the number of exclusion reads (i.e., reads that are mapped to exon-exon junctions that span a given segment). Reads mapped to multiple genomic locations were excluded from the analysis (i.e., only uniquely mapping reads were used). The percent spliced in (PSI, fraction of transcripts of a given gene that includes the exon) was calculated based on the inclusion and exclusion reads counts. Inclusion and exclusion read counts were modeled using the generalized linear model (GLM function in R) with a binomial distribution. Segments with FDR-corrected *p* < 0.05 (test for quasi-likelihood) were considered significant.

### 4.9. Transcripts Degradation Rate Measurement

To measure the degradation rates of NRAS PTC and NRAS total transcripts, ~1.5 × 10^5^ AML12 cells were seeded in 12-well plates and treated with actinomycin D (final concentration 5 µg/mL) for 1, 2, 3, 5, 6 or 9 h. After incubation, cells were harvested, and RNA was extracted with a TRIzol reagent (Thermo Fisher Scientific, Waltham, MA, USA) using the manufacturer’s protocol, followed by RT-qPCR analysis. To estimate the half-lives of the NRAS PTC and NRAS total transcripts, we used a nonlinear regression curve to fit experimental datapoints to the function y(x) = exp[−K × (x − x0)], where −K is an exponential decay, and x0 is the time offset. Nonlinear regression curves were plotted together with raw datapoints; standard deviation was calculated based on 3 replicates. For calculations, GraphPad Prism 6.0 software was used.

### 4.10. Formaldehyde-Crosslinked RNA-Immunoprecipitation (RIP)

Two 10-cm^2^ plates with ~5 × 10^6^ AML12 cells were used to prepare each sample (performed in two replicates). Cells were harvested, resuspended in 2 mL of phosphate-buffered saline (PBS) and crosslinked by adding 37% methanol-free formaldehyde (143 μL) and incubated for 10 min at room temperature. Crosslinking was terminated by the addition of 2-M glycine in water (685 μL). Cells were washed twice with ice-cold PBS, followed by centrifugation at 1000× *g* for 5 min at 4 °C. Cell pellets were resuspended in 1 mL of IP lysis buffer (50-mM HEPES (pH 7.5), 0.4-M NaCl, 1-mM EDTA, 1-mM DTT, 0.5% Triton X-100 and 10% glycerol), with an added 20 μL of 0.1-M phenylmethylsulfonyl fluoride (PMSF), 10 μL of a 100×Halt protease inhibitor cocktail (Thermo Fisher Scientific, Waltham, MA, USA) and 5 μL of a RiboLock RNase inhibitor (40 U/μL) (Thermo Fisher Scientific, Waltham, MA, USA). The lysates were sonicated (10 s ON, 10 s OFF, amplitude 20 μm, 10 cycles; QSonica sonicator, amplitude 20%) and centrifugated at 14,000× *g* for 3 min. Supernatant with crosslinked protein-RNA complexes was subjected to IP overnight at 4 °C with an anti-SFPQ (ab195352, Abcam, Cambridge, UK), anti-NONO (N8664, Sigma Aldrich, St. Louis, MO, USA), anti-DDX3 (A300-476A, Bethyl laboratories, Montgomery, TX, USA) or human IgG (control) antibody bound to preblocked Sepharose G-Beads (ab193259, Abcam, Cambridge, UK). Then, beads were 5× washed with the IP buffer, followed by a wash with the RIP buffer (50-mM HEPES (pH 7.5), 0.1-M NaCl, 5-mM EDTA, 10-mM DTT, 0.5% Triton X-100, 10% glycerol and 1% SDS). Samples were incubated at 70 °C for 1 h and centrifugated at 1000× *g* for 5 min. RNA samples were extracted using TRIzol following the manufacturer’s protocol and analyzed by reverse-transcription qPCR amplification using the primers listed in Appendix A.

### 4.11. Capture Hybridization Analysis of Morrbid lncRNA Targets (CHART) [24]

For cell lysate preparation, ~4 × 10^7^ AMl12 cells per each sample were washed with ice-cold PBS and then resuspended in a 1% paraformaldehyde solution in PBS. PFA crosslinking was done under agitation for 20 min at RT and quenched by adding 1/10 of the volume of 1.25-M glycine for 5 min at RT under agitation. Crosslinked cells were collected by centrifuging at 1000× *g* for 5 min and then twice-washed with PBS. Cells were lysed by adding 1 mL of WB100 buffer (100-mM NaCl, 10-mM HEPES (pH 7.5), 2-mM EDTA, 1-mM EGTA, 0.2% SDS and 0.1% N-lauroylsarcosine, supplemented with 1× protease inhibitors cocktail, 0.8-µM PMSF and 4 µL of RNase inhibitor (40 U/µL)) per 100 mg of tissue. The lysates were sonicated (30 s ON, 30 s OFF, at 20% amplitude, 5 cycles; sonicator QSonica, Newtown, CN, USA), followed by 10 µL of RNase inhibitor (40 U/µL), 5 µL of 1-M DDT in water and 5 µL of 100× protease inhibitors immediately after sonication.

For the CHART experiment, 250 µL of denaturant buffer (8-M urea; 200-mM NaCl; 100-mM HEPES (pH 7.5); 2% (*w/v*) SDS) and 750 µL of 2× hybridization buffer (1.5-M NaCl, 1.12-M urea, 10× Denhardt solution and 10-mM EDTA, pH 8) were added per 500 µL of lysate [24]. At this point, 100 µL from each sample were collected for the no-oligos control. For both the Morrbid CHART and for the control CHART, four biotinylated probes were used (Appendix A Biotinylated Probes used in CHART and RIP protocols). Fifty-four picomoles of each probe per 100 µL of extract were added to each sample and incubated at RT with gentle agitation for 7 h. After incubation, samples were centrifuged for 10 min at 16,000× *g* at RT, and the supernatant was transferred to a fresh tube supplemented with 50 µL of denaturant buffer and 100 µL of prerinsed MyOne Dynabeads Streptavidin C1 (65001, Thermo Fisher Scientific, Waltham, MA, USA). To allow a biotin-streptavidin interaction, samples were incubated overnight at room temperature with gentle agitation.

To minimize nonspecific binding, beads with Morrbid RNA-protein complexes were captured with the magnet and 5× washed with 1 mL of WB250 buffer. After the wash step, beads were captured with the magnet capture and resuspended in 30 µL of water. Samples were fractionated by SDS-PAGE; each Morrbid-specific band and corresponding gel position within the control lane were cut out of the gel and analyzed by mass spectrometry after the in-gel tryptic digestion of the proteins. Excised protein bands were cut into 1 × 1 × 1-mm cubes, transferred into sample tubes and destained with 50% acetonitrile (ACN) in 100-mM ammonium bicarbonate and dehydrated by the addition of 100% ACN. After ACN removal, samples were subjected to in-gel digestion by trypsin overnight at 37 °C. The digestion buffer solution contained 13-ng/μL Promega sequencing-grade modified trypsin in 10-mM ammonium bicarbonate containing 10% ACN. The resulting tryptic peptides were extracted from the gel by adding two volumes (in comparison to the digestion buffer solution) of 0.5% aqueous trifluoroacetic acid and incubating for 1 h. Then, an equal volume of ACN was added, and the samples were incubated for another hour. The samples were vacuum-dried and dissolved in a solution containing 3% ACN and 0.1% aqueous TFA before LC-MS/MS analyses.

For the LC-MS/MS analysis, peptides were separated on a 50-cm 75-µm inner diameter column packed in-house with Aeris Peptide XB-C18 2.6-µm resin (Phenomenex, Torrance, CA, USA). Reverse-phase chromatography was performed with an Ultimate 3000 Nano LC System (Thermo Fisher Scientific, Waltham, MA, USA), which was coupled to a QExactive HF benchtop Orbitrap mass spectrometer (Thermo Fisher Scientific, Waltham, MA, USA) via a nanoelectrospray source (Thermo Fisher Scientific, Waltham, MA, USA). The mobile phases were: (A) 0.1% (*v/v*) formic acid in water and (B) 0.1% (*v/v*) formic acid, 80% (*v/v*) acetonitrile and 19.9% (*v/v*) water. Samples were loaded onto a trapping column (100-µm internal diameter, 20-mm length and packed in-house with Aeris Peptide XB-C18 2.6-µm resin (Phenomenex, Torrance, CA, USA) in mobile phase A at the flow rate 6 µL/min for 5 min and eluted with a linear gradient of mobile phase B (5–45% B in 60 min) at a flow rate of 350 nL/min. Column temperature was kept at 40 °C. Peptides were analyzed on the QExactive HF benchtop Orbitrap mass spectrometer (Thermo Fisher Scientific, Waltham, MA, USA), with one full scan (300–1400 m/z, R = 60,000 at 200 m/z) at a target of 3e6 ions, followed by up to 15 data-dependent MS/MS scans with higher-energy collisional dissociation (HCD) (target 1e5 ions, max ion fill time 60 ms, isolation window 1.2 m/z, normalized collision energy (NCE) 28%, underfill ratio 2%) detected in the Orbitrap (R = 15,000 at fixed first mass 100 m/z). Other settings: charge exclusions—unassigned, 1 and >6; peptide match—preferred; exclude isotopes—on and dynamic exclusion—30 s was enabled. MS raw files were analyzed by PEAKS Studio 8.5 (Bioinformatics Solutions Inc., Canada) [58], and peak lists were searched against the Uniprot-Tremble FASTA (canonical and isoform) database version of March 2018 (84,951 entries) with methionine oxidation and asparagine and glutamine deamidation as the variable modifications. The false discovery rate was set to 0.01 for the peptide-spectrum matches and was determined by searching a reverse database. The enzyme specificity was set to trypsin in the database search. Peptide identification was performed with an allowed initial precursor mass deviation up to 10 ppm and an allowed fragment mass deviation of 0.05 Da.

### 4.12. Affinity Pulldown of Biotinylated RNA for the Detection of NRAS mRNA-Protein Complexes

To prepare the NRAS total and NRAS NMD minigenes, we performed cDNA synthesis and amplification of the total RNA from wild-type AML12 cells using Pfu DNA polymerase (EP0501, Thermo Fisher Scientific, Waltham, MA, USA) and primers from Appendix A. To synthesize NRAS minigenes, we used a forward primer containing the T7-promoter sequence at its 5′end and a gene-specific reverse primer; for control reverse NRAS minigenes, we used a gene-specific forward primer and reverse primer containing the T7-promoter sequence at its 3′end (Figure 3A). Two products of each PCR reaction—one corresponding to the NRAS total transcript and the second corresponding to the NRAS NMD transcript—were purified from the gel and confirmed by Sanger sequencing. To create the biotinylated NRAS minigenes, an in vitro T7 transcription was done using a Pierce RNA 3’ Desthiobiotinylation Kit, (20163, Thermo Fisher Scientific, Waltham, MA, USA) according to the manufacturer’s instructions, with the incubation time 4 h at 37 °C. The resulting biotinylated RNA products of the T7 transcription reaction were treated with DNase I to digest any traces of DNA and purified by 7.5% denaturing polyacrylamide gel (PAGE). Bands were excised, crashed and soaked in the buffer (500-mM sodium acetate, 89-mM Tris, 89-mM boric acid and 2-mM EDTA, pH 8.3) at 4 °C overnight; the supernatant was collected, and RNA was precipitated by adding 3 volumes of ethanol and isolated by centrifugation for 1 h at 10,000× *g*. Purified RNA was used to perform a RNA pull-down assay using a Pierce Magnetic RNA-Protein Pull-Down Kit (20164, Thermo Fisher Scientific, Waltham, MA, USA) according to the manufacturer’s instructions and then analyzed by RT-qPCR.

### 4.13. Western Blotting

Cell extracts were prepared in triplicates using ~10^6^ SFPQ knockdown or control cells after 6 days of SFPQ inhibition with siRNAs. Cells were lysed using a radioimmunoprecipitation assay buffer (RIPA) (Sigma Aldrich, St. Louis, MO, USA) supplied with a 100xHalt Protease Inhibitor Cocktail (Thermo Fisher Scientific, Waltham, MA, USA), according to the manufacturer’s protocol. Concentrations of the protein lysates were determined by the Bradford protein assay (Thermo Fisher Scientific, Waltham, MA, USA). Cell extracts (~30 µg) were denatured in Laemmli buffer (Bio-Rad Laboratories Inc., Hercules, CA, USA) by heating at 95 °C for 10 min and separated by electrophoresis in 10% SDS—polyacrylamide gel using PageRuler Prestained Protein Ladder (Thermo Fisher Scientific, Waltham, MA, USA) as the standard. Proteins were transferred to PVDF membranes (Bio-Rad Laboratories Inc., Hercules, CA, USA) using a Mini Trans-Blot Cell and Criterion Blotter (Bio-Rad Laboratories Inc., Hercules, CA, USA) at 80 V for 40 min at RT in the transfer buffer (25-mM Tris, 250-mM glycine (pH 8.3) and 10% ethanol). Then, PVDF membranes were blocked by incubation in a 0.05% Tween 20-TBS solution (10-mM Tris–HCl, pH 7.5 and 150-mM NaCl) with 5% bovine serum albumin (Sigma Aldrich, St. Louis, MO, USA) at 4 °C overnight. The blocked membrane was incubated with primary antibody anti-SFPQ SAB4200501 (Sigma Aldrich, St. Louis, MO, USA) 1:1000 dilution or anti-β-actin MA1-140 (Thermo Fisher Scientific, Waltham, MA, USA) 1:5000 dilution for 1 h at room temperature. After 3 washes with the 0.05% Tween 20-TBS solution, the appropriate secondary antibody was added for 1h: anti-rabbit IgG, HRP-linked antibody (7074P2, Cell Signaling Technology, Danvers, MA, USA); donkey anti-rabbit IgG H&L Alexa Fluor 680 (ab175772, Abcam, Cambridge, UK); anti-mouse Alexa 650. Membrane washed 3× with the 0.05% Tween 20-TBS solution and visualized in the ChemiDoc imager (Bio-Rad Laboratories Inc., Hercules, CA, USA). In the case of HRP-linked secondary antibodies using Clarity Western ECL blotting substrates (Bio-Rad Laboratories Inc., Hercules, CA, USA). Images were analyzed using ImageJ software.

### 4.14. Statistical Analysis of the Experimental Data

Most of the diagrams are based on at least three independent experiments. Statistical data processing was performed using the GraphPad Prism software (version 6) with a two-sample *t*-test. The data were considered statistically significant at *p* < 0.05.

## Figures and Tables

**Figure 1 ijms-21-05605-f001:**
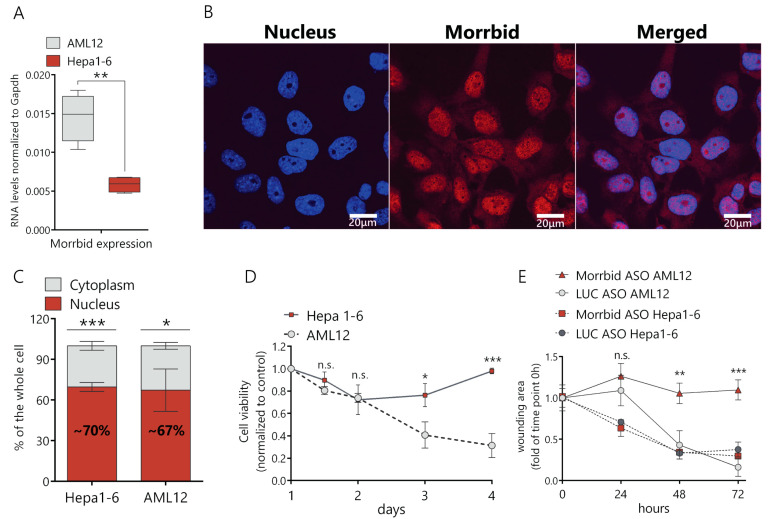
Characterization of Morrbid long noncoding RNA (lncRNA) expression, cellular localization and knockdown phenotype. (**A**) Comparison of Morrbid expression levels in AML12 normal murine hepatocytes and the Hepa1-6 hepatoma cell line with RT-qPCR. (**B**) Fluorescence in situ hybridization analysis (FISH) of Morrbid localization in AML12 cells (DNA was stained with Dapi and Morrbid was stained with a Cy5-labeled probe). (**C**) RT-qPCR analysis of Morrbid expression in the nuclear and cytoplasmic fractions extracted from AML12 cells. (**D**) Viability assay of Hepa1-6 and AML12 cells depleted in Morrbid, on the 1, 1.5, 2, 3 and 4 days of knockdown, normalized to control the luciferase antisense oligonucleotides (LUC ASO) treatment and viability at day 1 after the initial transfection. (**E**) Wound-healing assay of Morrbid knockdown (KD) and LUC control KD in AML12 and Hepa1-6 cells. The wound was introduced just after the initial transfection, and data were normalized to the wound area at the first timepoint. Results show mean ± SD. n.s.—not significant. * *p* < 0.05, ** *p* < 0.01 and *** *p* < 0.001.

**Figure 2 ijms-21-05605-f002:**
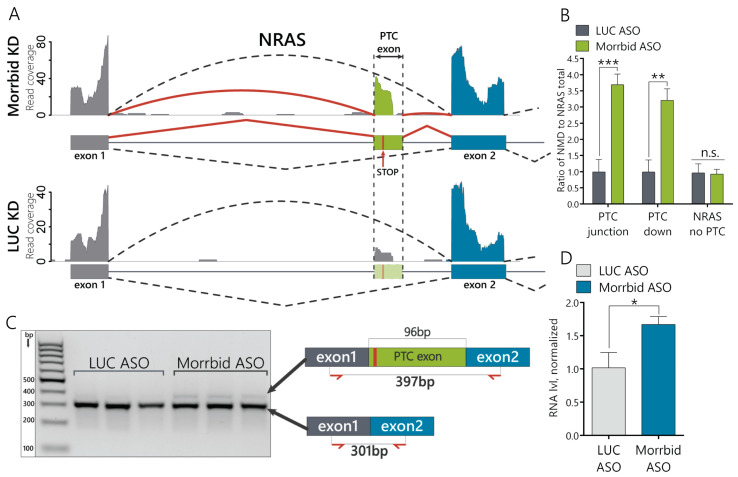
Morrbid lncRNA depletion leads to the enhanced incorporation of the NRAS premature termination codons (PTC) exon in murine hepatocytes. (**A**) RNA coverage of the zoomed region of the NRAS transcript in Morrbid and LUC control knockdowns. The solid and dashed arced lines represent the RNA coverage of the splice junctions. (**B**) RT-qPCR analysis of the NRAS isoform expression after Morrbid KD and control LUC KD using primers laying on the junction between the PTC exon and neighboring exons (PTC junction) and spanning across the junction to the downstream neighboring exon (PTC down). (**C**) Image of amplicon separation by agarose gel electrophoresis. Amplicons were obtained with primers spanning across the alternative NRAS exon to amplify the PTC and no PTC NRAS transcripts. (**D**) RT-qPCR analysis of the NRAS pre-mRNA level in the control LUC and Morrbid KD cells using primers that amplify the fragment with exon1 and intron. NMD: nonsense-mediated decay. Results show mean ± SD. n.s.—not significant. * *p* < 0.05, ** *p* < 0.01 and *** *p* < 0.001.

**Figure 3 ijms-21-05605-f003:**
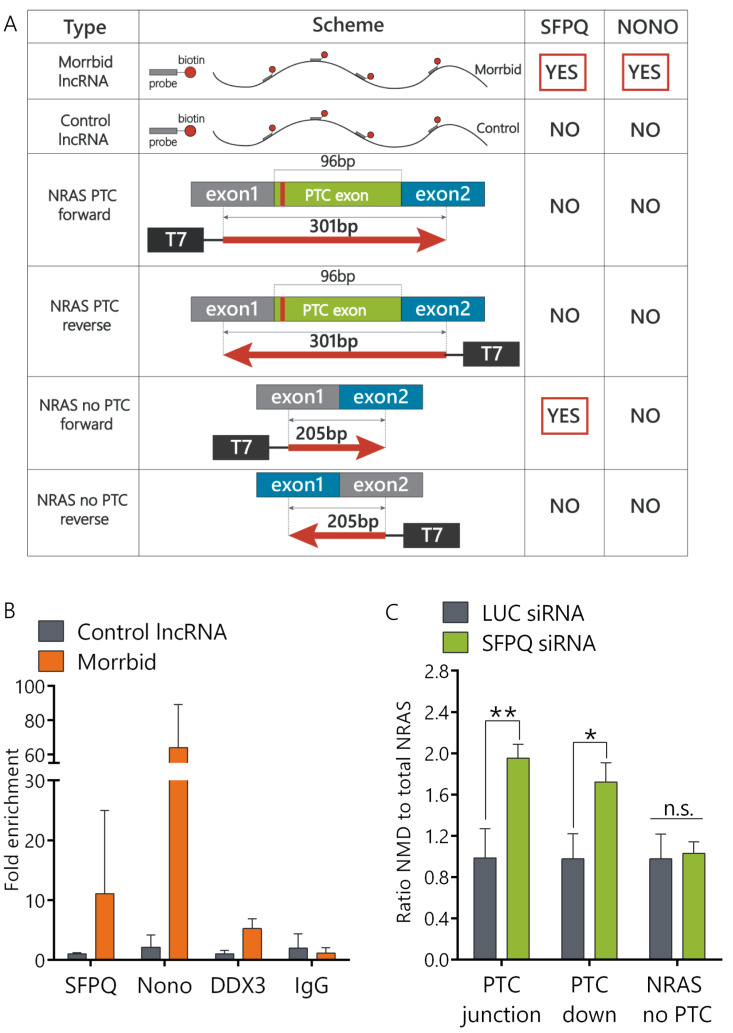
The SFPQ-NONO heterodimer interacts with Morrbid and influences NRAS PTC exon splicing. (**A**) Summary table of the Capture Hybridization Analysis of RNA Targets (CHART) and RNA pulldown assay results. (**B**) Fold enrichment of Morrbid lncRNA in the RNA immunoprecipitation assay (RIP) performed with SFPQ and NONO antibodies, as well as DDX3 and IgG antibodies as controls, quantified with RT-qPCR. (**C**) Relative expression of NRAS isoforms after 6 days of inhibition of the SFPQ protein (RT-qPCR analysis). siRNA: small interfering RNA. Results show mean ± SD. n.s.—not significant. * *p* < 0.05, ** *p* < 0.01.

**Figure 4 ijms-21-05605-f004:**
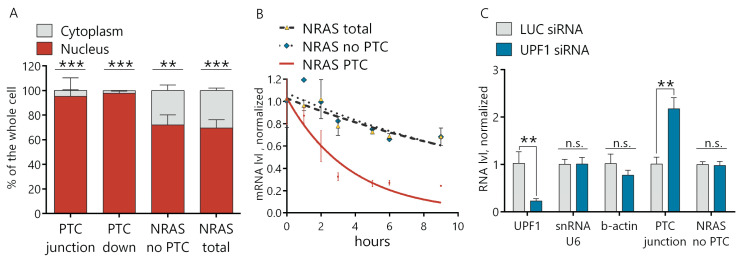
NRAS PTC transcript is degraded in the cytosol via the NMD pathway. (**A**) RT-qPCR analysis of NRAS PTC, no PTC transcripts and NRAS total in the nuclear and cytoplasmic fractions extracted from AML12 cells. (**B**) Estimation of the NRAS transcripts degradation rate by an actinomycin D assay. (**C**) RT-qPCR analysis of gene expressions after 6 days of knockdown of the UPF1 protein. Results show mean ± SD. n.s.—not significant. ** *p* < 0.01 and *** *p* < 0.001. mRNA: messenger RNA.

**Figure 5 ijms-21-05605-f005:**
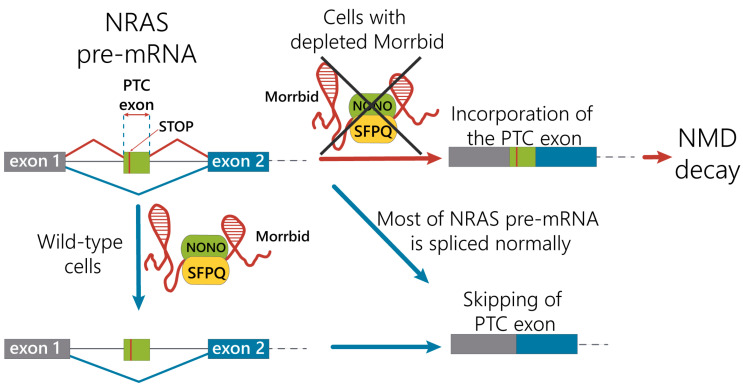
Proposed mechanism of the Morrbid lncRNA contribution in the regulation of NRAS mRNA alternative splicing.

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
