# Peer review of "Murine Long Noncoding RNA Morrbid Contributes in the Regulation of NRAS Splicing in Hepatocytes In Vitro"

_ijms, 2020, doi:10.3390/ijms21165605_

Round 1
Reviewer 1 Report
In the manuscript titled "Murine long noncoding RNA Morrbid contributes into regulation of NRAS splicing in hepatocytes in vitro " by Fefilova et al. the authors use various experimental approaches to study the functions of Morrbid lncRNA in hepatocytes. They found that Morrbid regulates proto-oncogene NRAS splicing preventing formation of transcripts with premature termination codon. They also demonstrated that these transcripts are degraded via the Nonsense Mediated Decay pathway. The experiments are thorough and well-justified. The findings are very interesting, in particular because they show different functions of the same lincRNA in different cell types. However, there are some inconsistencies and gaps in data interpretation that need to be addressed prior publication.
Major points.
- If NMD is active in morrbid KD cells, then why it does not eliminate an excess of NRAS PTC transcripts? (Fig. 2B). This should be discussed.
- In my opinion, the work contains two parts: Morrbid function in the of NRAS alternative splicing regulation and targeting NRAS PTC by NMD. It seems that these are two different mechanisms that prevent formation of NRAS aberrant transcripts, while their regulatory coupling looks speculative. I think these parts should be more clearly separated in the manuscript. In particular, it would be better to move section 2.3. to the end of the Results.
Other points.
- Line 263. “..while the expression of NRAS no PTC remained unchanged (Fig. 2G).” There is no Fig. 2G. I guess it has to be “4B”.
- Some important data is included in the supplementary section, for example Figure S4, that is inconvenient for the reader. At the discretion of the authors.
Author Response
We would like to thank you for your careful review of our manuscript. Your comments were highly relevant and contributed to improve the quality of this manuscript. The authors and I would like to thank you for the time you spent reviewing this manuscript. Below, you will find answers to your comments. In addition, we have highlighted the changes directly in the manuscript.
1. If NMD is active in morrbid KD cells, then why it does not eliminate an excess of NRAS PTC transcripts? (Fig. 2B). This should be discussed.
Thank you for the suggestion. We add this point in the discussion section.
2. In my opinion, the work contains two parts: Morrbid function in the of NRAS alternative splicing regulation and targeting NRAS PTC by NMD. It seems that these are two different mechanisms that prevent formation of NRAS aberrant transcripts, while their regulatory coupling looks speculative. I think these parts should be more clearly separated in the manuscript. In particular, it would be better to move section 2.3. to the end of the Results.
We agree with this viewpoint and moved section 2.3 to the end.
3. Line 263. “..while the expression of NRAS no PTC remained unchanged (Fig. 2G).” There is no Fig. 2G. I guess it has to be “4B”.
We corrected this point in the manuscript.
4. Some important data is included in the supplementary section, for example Figure S4, that is inconvenient for the reader. At the discretion of the authors.
Thank you for the suggestion. We included Fig. S4B in the main text of the manuscript.
Sincerely,
Dr. Olga Sergeeva
Reviewer 2 Report
The manuscript by Fefilova et al, "Murine long noncoding RNA Morrbid contributes into regulation of NRAS splicing in hepatocytes in vitro", gives insight into the mechanism of NRAS and role of lncRNA morrbid in regulation, is well written and to the point.
The introduction gives a brief and sufficient background.
The methods are adequately described.
The results are very clear.
Discussion stays in the realm of the presented data and results. It was good to see a manuscript claims and hypothesis are based on the results. There are many things that are expected from a study to reinforce the hypothesis, however, I am satisfied with the current scope of the manuscript.
I pleased to recommend this manuscript to the editor.
Author Response
The authors and I would like to thank you for the time you spent reviewing this manuscript and for the positive feedback.
Sincerely,
Dr. Olga Sergeeva
